# Extending the Host Range of Fusarium Poae Virus 1 from *Fusarium poae* to other *Fusarium* Species in the Field

**DOI:** 10.3390/v14102246

**Published:** 2022-10-13

**Authors:** Xiushi Song, Yidan Sun, Jing Gao, Kaixin Gu, Yiping Hou, Jianxin Wang, Mingguo Zhou

**Affiliations:** Key Laboratory of Pesticide, College of Plant Protection, Nanjing Agricultural University, Nanjing 210095, China

**Keywords:** Fusarium poae virus 1, *Fusarium asiaticum*, host transmission, hypovirulence

## Abstract

Fusarium poae virus 1 (FpV1, a betapartitivirus) is one of the mycoviruses which is discovered earlier. Due to the vegetative incompatibility barrier that often exists between different species or strains of filamentous fungi, FpV1 has been thought to be limited to its host, *F. poae*, as a non-hypovirulence mycovirus in the past 20 years in the field. Here, a novel strain of FpV1 (FpV1-Fa) with two dsRNA segments (2157-and 2080-nt) was consistently identified in *F. asiaticum* isolates in the field. FpV1-Fa induced abnormal morphology and hypovirulence of *F. asiaticum*, along with a high viral load. FpV1-Fa was detected only from the *F. asiaticum* and *F. tricinctum* strains at a FpV1-Fa sampling site (119.014289, 33.8261), while the other strains from other sites were not identified FpV1-Fa. A horizontal transmission experiment showed that FpV1-Fa can transfer from *F. asiaticum* to *F. poae* and *F. tricinctum*, but not to *F. graminearum*. The selection analysis of FpV1-Fa revealed RdRP and CP were under strong purifying selection, and the C-terminal side of RdRP was under positive selection. In these regions, 9 amino acid mutations in RdRP and 21 mutations in CP appeared to cause the variation of host range and virulence in FpV1-Fa.

## 1. Introduction

*Fusarium* is a globally important genus of fungal pathogens which is responsible for devastating diseases of plants and serious diseases of humans, such as Fusarium head blight (FHB) and keratitis. Among the *Fusarium* species causing FHB, *F. graminearum* and *F. asiaticum* are the dominant species isolated in China (*F. asiaticum* previously referred to as *F. graminearum* [1,2]), although *F. poae* has been found in diverse substrates such as barley and wheat [3,4]. In recent years, a variation in the composition of the FHB prominent species has been observed. In particular, *F. avenaceum* and *F. poae* have increased their frequencies, while *F. graminearum* and *F. culmorum* showed a reduced presence [5,6,7]. In Argentina, *F. graminearum* and *F. poae* predominate over other *Fusarium* species [8]. This change may be caused by climatic factors. Covarelli et al. show that *F. poae* increases its presence when the climatic conditions are not suitable for *F. graminearum* growth [5].

Several RNA mycoviruses have been discovered from *F. asiaticum* and *F. graminearum*. According to the tenth report of the International Committee for Taxonomy of Viruses (ICTV), these mycoviruses were assigned to six families and one unassigned dsRNA group, including *Totiviridae* (FaVV1), *Chrysoviridae* (FgV-ch9 and FgV2), *Fusagraviridae* (FgV3), *Hypoviridae* (FgHV1 and FgHV2), *Fusariviridae* (FgV1), *Tymoviridae* (FgMTV1), and the Unassigned group (FgV4 and FgV5, possibly related to the families *Amalgaviridae* and *Partitiviridae*) [9,10]. The natural host ranges of mycoviruses are thought to be limited to a single species or to very closely related species because the only proven route of mycovirus horizontal transmission in filamentous fungi is through hyphal fusion [11]. Although incompatibility has long been known to restrict virus transmission, diverse mycoviruses have been experimentally transferred into fungal species different from the original host via transformation by infectious viral cDNA [12], transfection by infectious virions [13], in vitro-synthesized viral transcripts [14], coculturing [15,16], and protoplast fusion [17,18]. Fusarium poae virus 1 (FpV1) is a *Partitiviridae* virus that is first isolated from the *F. poae* strain A-11 [19]. The FpV1 genome contains two dsRNAs, 2185 and 2203 bp in size. The two dsRNAs encode the viral RdRP and CP, 70 and 74 kDa in size, respectively. FpV1 is stable even after repeated subculturing but does not seem to induce abnormal morphology or changes to virulence in *F. poae* [9]. Research conducted in the past 20 years or so has proven the natural host ranges of FpV1 were thought to be limited to *F. poae* until the FpV1-like mycovirus (FpV1Monilinia-TNS) was found in *Monilinia fructicola* [20]. It is puzzling because there is no report of isolating FpV1 from *Fusarium* even though the relatedness of *F. poae* to *Fusarium* spp. is more closely to that of *M. fructicola*.

The mycovirus research is of great interest for their potential as an effective tool for biocontrol of fungal pathogens [21]. However, two major challenges limit the use of mycoviruses as a biocontrol agent against phytopathogens. One problem is that only a few isolates have a hypovirulent effect on their host [10], and another one is the limited transmission and stability [22,23,24]. Here, a novel strain of FpV1 (FpV1-Fa), which shared 90.5% of nucleotide identities with previously reported FpV1, was consistently identified in *F. asiaticum* and *F. tricinctum* isolates (mean prevalence: 13%) that were collected in Jiangsu, China. It is worth noting that the sequence similarity between FpV1-Fa and FpV1Monilinia-TNS was higher than that between FpV1-Fa and FpV1. The analysis of transmission and host range of FpV1-Fa showed that the novel FpV1-Fa can infect *F. poae*, *F. tricinctum* and *F. asiaticum* with high stability. In *F. asiaticum*, FpV1-Fa induced abnormal morphology and decreased virulence. The selection analysis of FpV1-Fa revealed that RdRP and CP were under strong purifying selection, and the C-terminal side of RdRP was under positive selection. These selective sites appeared to cause differences in morphology and transmission of FpV1-Fa and FpV1.

## 2. Materials and Methods

### 2.1. Strains, Culture Conditions, and Biological Characterization

During 2019 and 2021, 339 *Fusarium* strains (Table 1) were isolated from scabby wheat spikes collected from different Fusarium head blight epidemic regions in different areas of China. To obtain monoconidial strains, infected scabby grains were washed with sterile water. A 20 µL aliquot was smeared onto water agar (WA) plates amended with 50 µg mL^−1^ penicillin and incubated at 25 °C overnight. A germinated conidium of *Fusarium* was recovered for each individual sample and transferred to a new plate containing penicillin to help inhibit bacterial growth. For regeneration, fungal strains were cultured on potato dextrose agar (PDA). The species of *Fusarium* spp. strains were identified by sequencing the translation elongation factor-1alpha (EF-1α) gene. Genomic DNAs were extracted using a DNA extraction kit (Transgen, Beijing, China) and then amplified with EF-1α primer pair (EF-1α-F 5′-ATGGGTAAGGARGACAAGAC and EF-1α-R 5′-GGARGTACCAGTSATCATGTT).

### 2.2. RNA Sequencing

Each of the 25 isolates was cultured on a PDA plate for 7 days. Total RNA was extracted from the one-gram mycelial mass using Trizol reagent (Invitrogen, Carlsbad, CA, USA) according to the manufacturer’s instructions. Total RNA (2 μg) from each of the 25 isolates was mixed for RNA-Seq analysis. Sequencing was carried out by Novogene Bioinformatics Technology Co. Ltd. Beijing, China, using an Illumina HiSeq 2000 instrument. Contigs were obtained and subjected to local BLAST against a nucleotide database using BLASTn.

### 2.3. dsRNA Extraction, Purification, cDNA Cloning and Sequencing

Mycelial plugs of *Fusarium* strains were cultured on PDA plates overlaid with cellophane membranes for 5 days at 25 °C in the darkness. Viral dsRNA from *Fusarium* strains were extracted by cellulose (Sigma, Dorset, England) chromatography. DNase Ι and S1 nuclease (TaKaRa, Shiga, Japan) were used to eliminate DNA and ssRNA contamination according to the manufacturer’s instructions. M-MLV Reverse Transcriptase (Promega, Madison, WI, USA) and PrimeSTAR^®^ HS DNA polymerase (TaKaRa, Shiga, Japan) were used for cDNA synthesis and PCR in a Thermal Cycler (TaKaRa TP600, Dalian, China) based on the protocol, respectively. Specific primers were designed to amplify virus sequences (Appendix A).

### 2.4. Semi-Quantitative RT-PCR

The semi-quantitative RT-PCR was carried out in a volume of 20 µL containing 2 µL 10 × PCR buffer, 0.24 µL DNA polymerase, 1.6 µL dNTP (2.5 mM), 2.4 µL each primer, 1.3 ul cDNA, and made up to 20 µL with water. Gene-specific primers (RdRP-F/RdRP-R) were designed using Primer Premier5 software (PREMIER Biosoft International, Palo Alto, CA, USA). PCR amplification was performed under the following conditions: 95 °C for 90 s, followed by 30 cycles of 94 °C for 15 s, 54 °C for 30 s, 72 °C for 20 s, and the final extension at 72 °C for 5 min. To normalize the total amounts of cDNAs present in each reaction and to eliminate the differences among the samples, the *α-tubulin* housekeeping gene was co-amplified as an internal control. The PCR products were electrophoresed in the 1% agarose gels with 0.6 mg/mL ethidium bromide.

### 2.5. Phylogenetic Analysis

Sequences were aligned with the Clustal W program. Sequence similarity searches were performed in NCBI BLAST program (Blastn, Blastp). The construction of the phylogenetic tree was carried out with the neighbor joining method in MEGA 6 program [25].

### 2.6. Generation of Isogenic Fusarium Free of Mycovirus

Four methods were used to cure *Fusarium* of FpV1 as previously described, including cold treatment, temperature shock, hyphal tipping and antibiotic treatment [20]. Virus-free lines were confirmed by RT-PCR with species-specific primers (Appendix A).

### 2.7. Mycelium Growth, Conidiation Formation, and Pathogenicity Assays

Mycelial agar plugs of the *Fusarium* strains were transferred to PDA in Petri dishes (9 cm diameter). The diameter of each colony was measured after 3 days, and the diameter difference between the two measurements was used to calculate the radial mycelial growth rates. For conidiation assays using conidia as initial inoculums, 10 μL of macroconidia (5 × 10^5^ mL^−1^) of each strain was transferred to 20 mL CMC in a 50 mL conical flask and incubated at 28 °C under constant fluorescent lighting on a shaker (200 rpm) and the number of conidia was calculated at day 3. The pathogenicity assay was performed using seedling inoculation. Three−day−old wheat seedlings were inoculated with 3 µL of macroconidia suspension (5 × 10^5^ spores/mL), and seedling inoculations were carried out in growth chambers [26]. Thirty seedlings were inoculated, and the brown lesions of diseased seedlings were measured at seven days post inoculation (dpi).

### 2.8. Virus Transmission

The hyphal anastomosis method was used to transmit the virus from an infected fungal strain into a virus-free *F. asiaticum* strain 2021-HygB (hygromycin-resistant), which was constructed by *Agrobacterium tumefaciens*-mediated transformation [27]. A *PLS1* gene (FG08695) encoding a tetraspanin that is dispensable in *F. asiaticum*, served as the target site for integration of the hygromycin resistance gene [27]. *F. tricinctum* and *F. poae* strains that had a hygromycin B or G418 resistance gene were constructed by protoplast transformation [27]. During hyphal anastomosis, 2021-HygB was used as the recipient and the virus-infected HA 3-11 strain was used as the donor. The donor and recipient strains were inoculated together on the same PDA plate and incubated at 25 °C for 5 days. Mycelial plugs were taken from the growth side of 2021-HygB and transferred on a PDA plate containing 200 µg/mL hygromycin B (Coolaber, Beijing, China). We collected the mycelial plugs that grew on the PDA plate containing hygromycin B and then isolated single spores. The spores were examined with dsRNA extraction and RT-PCR process.

### 2.9. Selection Analysis

Selection analysis was conducted using a sliding window analysis of K_a_ and K_s_ (SWAKK) [28]. The SWAKK 2.1 webserver was used to calculate the K_a_/K_s_ ratio across the gene alignment using a sliding window of 10 codons.

## 3. Results

### 3.1. Fusarium Species Determination, Sequencing Analysis and Virus Assays

During the period 2019–2021, a total of 107 *Fusarium* isolates were recovered from rachis tissue affected by FHB collected from Huai’an of Jiangsu Province, China, and were identified by amplifying fragments of translation elongation factor gene (*TEF1*). The evolutionary history of 43 *Fusarium* strains isolated in 2021 was studied at the genetic level by sequencing fragments of the internal transcribed spacer region (ITS). A phylogenetic analysis of the concatenated sequences of about 530 nt ITS was carried out. The phylogenetic tree showed that there were four separated clades corresponding to *F. graminearum*, *F. asiaticum*, *F. tricinctum,* and *M. nivale*. *F. asiaticum* was the most common species of the FHB complex, comprising 65% of the total isolates. *F. graminearum* came second, comprising 26% of FHB complex species. The percentage of isolates identified as *F. tricinctum* 7% and *M. nivale* 2% at the sampling site (Figure 1A).

Total RNAs were extracted from the 43 *Fusarium* isolates and a major dsRNA segment was detected through electrophoresis with the size of approximately 2.2 kb after DNase I and S1 nuclease digestion. Five isolates contained the 2.2 kb dsRNA segments, including three *F. asiaticum* strains and two *F. tricinctum* strains. Two contigs representing the complete genomic segments of a bipartite virus were identified by RNA sequencing (Appendix A). The sequence shared the highest sequence identity with Fusarium poae virus 1 (FpV1) (genus *Partitivirus*, family *Partitviridae*), a double-stranded RNA virus. So, the isolates here from *F. asiaticum* and *F. tricinctum* were designated as Fusarium poae virus 1 isolate FpV1-Fa. One segment (RNA1, 90.5% identities with FpV1 RNA2) was 2157 nt, with a GC content of 43.72%, encoding an RdRP-like protein of 673 amino acid residues with an estimated mass of 78.18 kDa. The 5′UTR extended from nucleotide 1 to 57 and the 3′UTR from nucleotide 2080 to 2157. BLAST search indicated that the deduced protein contains a reverse transcriptase-like family (RT-like superfamily) (Figure 1B). The RNA2 segment (83.98% identities with FpV1 RNA1) was 2080 nucleotides in length. The ORF encoded a coat protein (CP) of 637 amino acid residues in length, with an estimated molecular weight of 70.73 kDa, and two short untranslated regions (UTRs) of 101 nt and 65 nt in length at the 5′- and 3′-terminus, respectively (Figure 1B).

### 3.2. Incidence and Distribution of FpV1-Fa

To investigate the incidence and distribution of FpV1-Fa in China, 339 *Fusaium* strains were tested for the presence of FpV1-Fa by using RT-PCR with the primer pair RdRP-F/RdRP-R (Appendix A). These strains were isolated from Huai’an around the FpV1-Fa sampling site and other provinces in China (Appendix A). Fourteen isolates out of 107 *Fusarium* strains (13%) which isolated from Huai’an of Jiangsu Province contained FpV1-Fa. FpV1-Fa infection was detected only in the *F. asiaticum* and *F. tricinctum* strains from the FpV1-Fa sampling site (119.014289, 33.8261), while the other strains from other sites were not identified FpV1-Fa, even their distribution was close to the site of FpV1-Fa (Table 1).

### 3.3. Viral Transmission and Stability

Horizontal transmission of FpV1-Fa from virus-transfected strains to virus-free strains was examined by dual culture on PDA. Mycelial plugs of virus-infected donor strains (HA3-11) and of virus-free, recipient strain (Fa2021-HygB, which had a hygromycin B resistance gene) were placed on each PDA plate. After certain periods of coculture, mycelial plugs were taken from three different positions (F, far from border; M, middle; N, near border) on recipient sides and sub-cultured onto new PDA plates, followed by dsRNA extraction and RT-PCR test by using the primer pair RdRP-F/RdRP-R (Figure 2A). Results showed that the recipient strain Fa2021-HygB contained FpV1-Fa dsRNA with a 100% infection rate from the N agar plugs, a 66.7% infection rate from the M agar plugs, and a 40% infection rate from the F agar plugs (Figure 2B). We also attempted to transfer FpV1-Fa from *F. asiaticum* to *F. poae*, *F. graminearum*, *F. tricinctum*, *Magnaporthe oryzae*, and *Sclerotinia minor*. Consequently, the RT-PCR showed that FpV1-Fa was transmitted to *F. poae* and *F. tricinctum* with a 100% infection rate from the N agar plugs. However, no specific PCR product was obtained from other sites where the agar plugs had been taken from (M and F mycelia of *F. poae* and *F. tricinctum*; N, M, and F mycelia of *M. oryzae* and *S. minor*) (Figure 2B). To investigate the accumulated level in the recipient strains, semi-quantitative RT-PCR analysis was carried out with a specific primer pair (Appendix A). The *α-tubulin* gene was used to balance the overall RNA amount. In the recipient strain Fa2021-HygB, the accumulation of FpV1-Fa decreased continuously from near border to far border, compared to that in donor strain HA3-11. FpV1-Fa accumulated in the recipient Ft-HygB (*F. tricinctum* strain that had a hygromycin B resistance gene) at a very low level but at a high level in the Fp-Neo (*F. poae* strain that had a G418 resistance gene) (Figure 2C).

We also examined the elimination of FpV1-Fa through conidia, cold treatment and hyphal tipping with and without ribavirin. FpV1-Fa was easily transmitted to single-conidial isolates in natural *F. asiaticum* strain HA3-11 and HA5-95 with a 100% poisoned rate of conidia. We further analyzed the stability of FpV1-Fa transmission in HA3-11 using secondary single-conidial isolation and hyphal tip isolation. Thirty secondary single-conidial isolates from the FpV1-Fa-positive primary single-conidial isolate contained viral dsRNA in all isolates (Figure 2D). The cold treatment and hyphal tipping with ribavirin methods also failed to eliminate FpV1-Fa in HA3-11 (Figure 2D). These results indicated FpV1-Fa was stable in *F. asiaticum*.

### 3.4. Impact of FpV1-Fa on Host Biological Properties

We then detected the morphological differences between the dsRNA-free strain and the dsRNA-containing strain. After culturing for 3 days, like strain HA3-11, the FpV1-Fa recipient strain of Fa2021-HygB-Fpv1Fa had reduced growth rate and dense hyphae relative to virus-free strains Fa2021-HygB. The growth rate of Fa2021-HygB-Fpv1Fa was reduced by 28% compared to Fa2021-HygB (Figure 3A). There was no significant difference (*p* > 0.05) in the production of conidia between Fa2021-HygB-Fpv1Fa and Fa2021-HygB, though strain HA3-11 produced fewer macroconidia. However, the morphology of macroconidium was affected by FpV1-Fa. The macroconidia were shorter in Fa2021-HygB-Fpv1Fa and HA3-11 strains than in Fa2021-HygB. The ratio of macroconidia larger than 20 μm was reduced by 59% for Fa2021-HygB-Fpv1Fa relative to Fa2021-HygB (Figure 3B). The virulence of the FpV1-Fa recipient strain was assayed by seedling inoculation with conidial spores. Seedling inoculation assays revealed that lesion length was reduced by 62% for strain Fa2021-HygB-Fpv1Fa, compared to strain Fa2021-HygB at 7 dpi (Figure 3C).

### 3.5. Phylogenetic and Selection Analysis of FpV1-Fa

To define the phylogenetic relationship of FpV1-Fa with other viruses in *Fusarium* species, a phylogenetic tree was established based on the RdRP domain. FpV1-Fa firstly formed a tight cluster with FpV1Monilinia-TNS2 from *Monilinia* and then clustered with mycoviruses FpV1 from *F. poae*, forming an independent clade of *Partitiviridae* (Figure 4A). The phylogenetic tree based on the CP sequence was similar to the above result, in which FpV1-Fa clustered with mycoviruses of *Partitiviridae* (Figure 4B). To assess how mutations and phenotypes align with selective pressures, we plotted this analysis for CP and RdRP by applying a sliding window analysis of K_a_ and K_s_ (SWAKK) to assess the selective pressure on CP and RdRP (Figure 4C). SWAKK calculates the ratio of non-synonymous to synonymous substitution rates (K_a_/K_s_) in a pairwise alignment. Codons with ratios below one are considered under purifying selection, and those above one are under positive selection. For RdRP_FpV1-Fa_, relative to other RdRP_FpV1_, RdRP_FpV1-240374_ and RdRP_FpPV12516_, most of the mutant sites were in areas of purifying selection (Figure 4C). The strongest purifying selection was surrounding the codons from 32 to 53, 75 to 97, 167 to 230, 300 to 336, and 408 to 504. Just the sites near the C-terminal side of RdRP were under strong positive selection, where adaptive molecular evolution should be identified. There were some areas of positive selection when analyzing pairwise nucleotide alignments relative to RdRP_FpV1Monilinia-TNS2_, such as codons from 138 to 140, 371, 543 to 550, and 591, indicating RdRP was under diversifying selection between FpV1-Fa and FpV1Monilinia-TNS2. For CP_FpV1-Fa_, relative to CP_FpV1_, CP_FpV1-240374_ and CP_FpPV12516_, all codons were in areas of purifying selection, suggesting the nonsynonymous mutations of CP were deleterious and were fixed at a lower rate than synonymous mutations (Figure 4D), especially codons from 23 to 104, 138 to 173, 248 to 277, 296 to 304, 334 to 352, 436 to 494, 523 to 538, 551 to 580, and 617 to 632. However, compared to CP_FpV1Monilinia-TNS2_, codons from 377 to 380, 506 to 511, and 595 to 609 were under positive selection, indicating nonsynonymous mutations were beneficial and favored by natural selection.

Given that the host range expanding and growth inhibiting of FpV1-Fa was caused by point mutations in CP and RdRP genes, we analyzed the natural mutant sites under positive and purifying selection. There were 9 amino acid sites of RdRP identical in FpV1-Fa and FpV1Monilinia-TNS2, but different in FpV1, FpPV12516 and FpV1-240374 (Figure 4E, marked in orange). Eleven different sites changed independently in FpV1-Fa or FpV1Monilinia-TNS2 (Figure 4E, marked in green). For CP protein, 21 amino acid sites varied both in FpV1-Fa and FpV1Monilinia-TNS2 (for example 26 L (I), Figure 4E), and 3 sites varied independently in FpV1-Fa or FpV1 Monilinia-TNS2, compared to the amino acid sequence of FpV1, FpPV12516, and FpV1-240374. The nine amino acid mutations of RdRP and 21 amino acid mutations of CP were thought to cause the host range expanding and growth inhibiting of FpV1-Fa.

## 4. Discussion

Viruses infect virtually all forms of cellular life, including animals, plants and fungi [29]. In some cases, mycoviruses confer a hypovirulent phenotype, which reduces the growth rate of their host and/or reduces their virulence [30,31,32]. Mycovirus research has been stimulated by the idea that they could be an effective tool for biocontrol of fungal pathogens [33]. To properly weaponize mycoviruses as biocontrol agents, a better understanding of their basic biology, including transmission modes and molecular mechanisms of parasitism, is needed [29]. *Fusarium* is an important genus of plant pathogenic fungi, and is widely distributed in soil and associated with plants worldwide [34]. Although the diversity of mycoviruses in *Fusarium* is increasing continuously due to the development of RNA deep sequencing techniques [10], only a few isolates have a hypovirulent effect on their host, which is one of the major challenges to using mycoviruses efficiently as a biocontrol agent against *Fusarium* [35]. Fusarium poae virus 1 (FpV1), is one of the mycoviruses that is discovered earlier and studied well [19,21,36,37]. FpV1 genome consists of two dsRNAs, 2185 and 2203 bp in size, encoding CP and RdRP, respectively [19]. Research conducted in the past 20 years or so has proven the natural host ranges of FpV1 are thought to be limited to *F. poae* until the FpV1-like mycovirus (FpV1Monilinia-TNS) is found in *Monilinia fructicola* [19,20,21,37]. Here, the genome of FpV1-Fa is highly similar to FpV1 Monilinia-TNS, but is different from the genome of the three FpV1 strains reported previously. Meanwhile, FpV1-Fa obtains a wide range of hosts, including *F. asiaticum*, *F. tricinctum*, and *F. poae*. Another important difference between FpV1-Fa and other FpV1 viruses is that FpV1-Fa retards *F. asiaticum* growth and virulence, while all the FpV1s including FpV1 Monilinia-TNS do not seem to induce abnormal morphology or pathogenicity, which is a common observance for *partitiviruses* generally.

Researchers believe that host range is determined by virus intrinsic factors, such as genetic traits determining its fitness in different hosts [38]. Experimental analyses have shown the relevance in host range evolution of across-host fitness tradeoffs [38]. Compared to the genome of FpV1, FpV1-Fa changed the length of 5′ and 3′UTR in both *RdRP* and *CP*. We are not sure if the changes of the 5′ and 3′UTR in FpV1-Fa will result in host shifting. However, it was reported that the 5′ end of RNA conveys important information on virus self-identity. For the yeast L-A virus, the 5′ diphosphates are essential to acquire a cap structure from host mRNA by a cap-snatching mechanism, the 5′ diphosphates of the strands have prominent roles during the viral replication cycle [39,40]. Host range is also determined by ecological factors extrinsic to the virus, such as the distribution, abundance, and interaction of species [38]. In our study, FpV1-Fa was only identified from Huai’an, China, indicating FpV1-Fa does not appear to spread. Given that RdRP and CP genes of FpV1-Fa shared with 95.48% and 93.87% identity with that of FpV1Monilinia-TNS which was isolated from Western Australia, the emerging of highly homologous strains in different regions spanning a broad area is still a mystery. One other thing to note is that the viral transmission and abundance of FpV1-Fa were different in *Fusarium* species. Like all viruses, mycoviruses are dependent on the host for replication, transcription, and translation [41]. Mycoviruses usually appear in the cytoplasm of their hosts, and they are generally persistent and vertically transmitted, passing through cell divisions [21]. It has been suggested that the viral titer and variability in the host spectrum are linked with the presence or absence of some host range genes (*Poxviruses*) [42]. The genomes of *F. asiaticum* and *F. graminearum* are similar, but why FpV1-Fa easily infects *F. asiaticum* but not *F. graminearum* should be investigated to uncover the mechanism of interaction between FpV1-Fa and *Fusarium*.

FpV1-Fa had an inhibitory effect on *F. asiaticum* growth. Meanwhile, it had a high viral load in *F. asiaticum.* The research on mycoviruses *F. oxysporum* f. sp. dianthimycovirus 1, Heterobasidion partitivirus 13 strain an1, Phytophthora endornavirus 2 (PEV2), and PEV3 show that the high titer of mycoviruses can induce more severe symptoms in the host than mycoviruses with low titer [43,44,45]. Some researchers have hypothesized that hypovirulence-associated mycoviruses causing the hypovirulence of hosts result from the codon usage of mycovirus and host was similar which was beneficial to increase mycovirus accumulation [46]. In some cases that may be true, but here the codon usage was not the key reason. In many cases, mycoviruses are known to reduce the growth rate of their host and/or reduce their virulence. This observation, however, creates a paradox as most mycoviruses are predominately transmitted vertically, which, according to theoretical predictions, should select for more mutualistic interactions [29]. It is possible, therefore, to increase the mutations helpful to mycovirus horizontal transmission would break the paradox.

Selection analyses using a sliding window analysis indicated that RdRP and CP were under a strong purifying selection, but except for C-terminal side of RdRP. RNA viruses usually form populations with high genetic variation. Such viral populations, which are known as quasispecies, maintain the balance between the continuous generation of mutations and the natural selection that acts on the mutants in relation to their fitness [47]. Among plant RNA viruses, the tobacco etch virus experienced a 5% decline in fitness per passage for up to 11 passages [48]. Although a few lineages experienced an increase in fitness, fitness decline has been the dominant phenomenon in various experiments with RNA viruses [49]. Such sensitivity of RNA viruses to deleterious mutations suggests that the fitness of RNA viruses would be dominated by purifying selection of deleterious mutations. Many amino acids were varied in areas of purifying selection both in RdRP and CP proteins. The variation of those amino acids to FpV1-Fa host range and virulence should be investigated carefully.

## Figures and Tables

**Figure 1 viruses-14-02246-f001:**
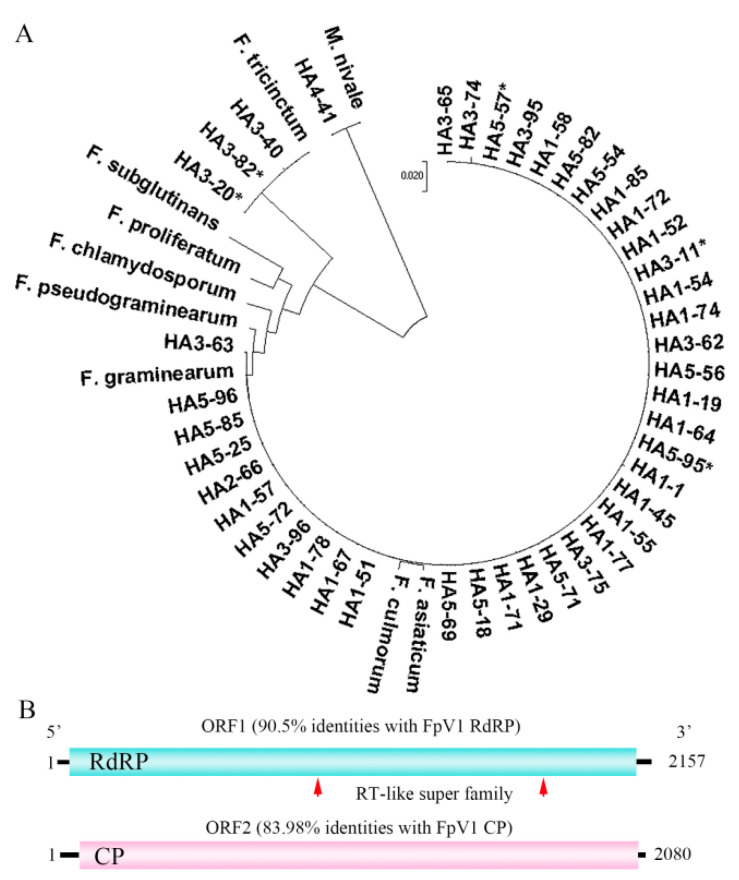
Phylogenetic analysis of *Fusarium* strains that isolated from Huai’an in 2021 and genomic organization of FpV1-Fa virus. (**A**) A phylogenetic analysis of the concatenated sequences of about 530 nt ITS is performed with the use of NJ method in the MEGA 6.0 program. The scale bar represents a genetic distance. The GenBank accession numbers of reference sequences are as follows: KX066046.1 (*F. graminearum*), NR_121320.1 (*F. asiaticum*), LC13674 (*F. proliferatum*), LC13682 (*F. subglutinans*), BBA62170 (*F. chlamydosporum*), NRRL28062 (*F. pseudograminearum*), EU214562.1 (*F. culmorum*), KU350729.1 (*F. tricinctum*) and MG891797.1 (*Microdochium nivale*). The isolates containing FpV1-Fa are shown with asterisks. (**B**) Genomic organization of FpV1-Fa virus. The open reading frame (ORF) and the untranslated regions (UTRs) are indicated by a rectangular box and a single line, respectively. The identities between Fusarium poae virus 1 (FpV1) and FpV1-Fa are marked on top.

**Figure 2 viruses-14-02246-f002:**
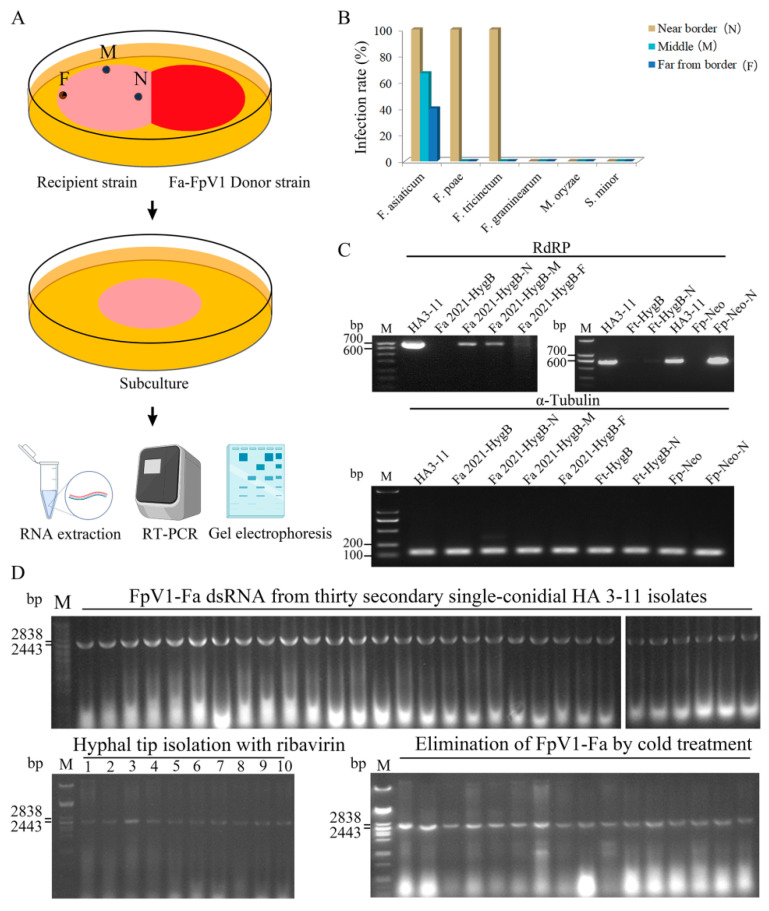
Viral transmission and stability analysis of FpV1-Fa. (**A**) Experimental procedure for investigating the horizontal transfer of FpV1-Fa from *F. asiaticum* to other fungi. F, far from border; M, middle; N, near border. (**B**) RT-PCR detection of FpV1-Fa infection from *F. asiaticum* to *F. poae*, *F. graminearum*, *F. tricinctum*, *Magnaporthe oryzae* and *Sclerotinia minor*. (**C**) The semi-quantitative RT-PCR detection of viral load in *F. asiaticum* and *F. tricinctum.* The α-tubulin gene is used to balance the overall RNA amount. (**D**) RT-PCR detection of the elimination of FpV1-Fa through conidia, cold treatment, and hyphal tipping with and without ribavirin. M, DNA marker.

**Figure 3 viruses-14-02246-f003:**
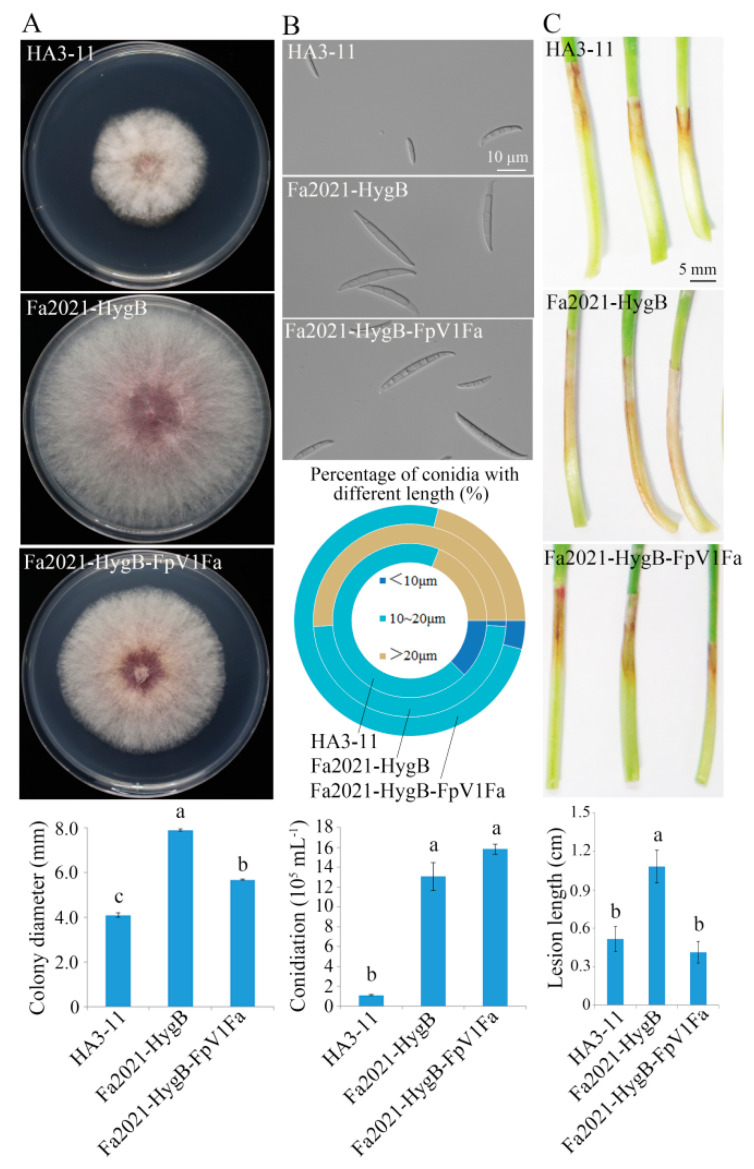
Impact of FpV1-Fa on *F. asiaticum* biological properties. (**A**) Colony morphology and growth rate of virus-free and -infected *F. asiaticum* strains 3 days after inoculation. Mycelial plugs are inoculated and cultured on PDA for 3 days at 25 °C in the darkness. The experiment is repeated three times. Values are means ± SD of three independent biological duplicates. (**B**) Macroconidia morphology and conidiation of virus-free and -infected *F. asiaticum* strains 5 days after inoculation. Lengths of conidia produced in CMC medium after 5 d. A total of 100 conidia is examined to detect the length and septa number for each strain. Experiments are performed in triplicate. (**C**) The virulence of virus−free and −infected *F. asiaticum* strains on wheat seedlings, as indicated by lesion length. Lesions are measured at 7 dpi and values are means ± SD of 30 replicate seedlings. Means with different letters are significantly different at *p* < 0.05.

**Figure 4 viruses-14-02246-f004:**
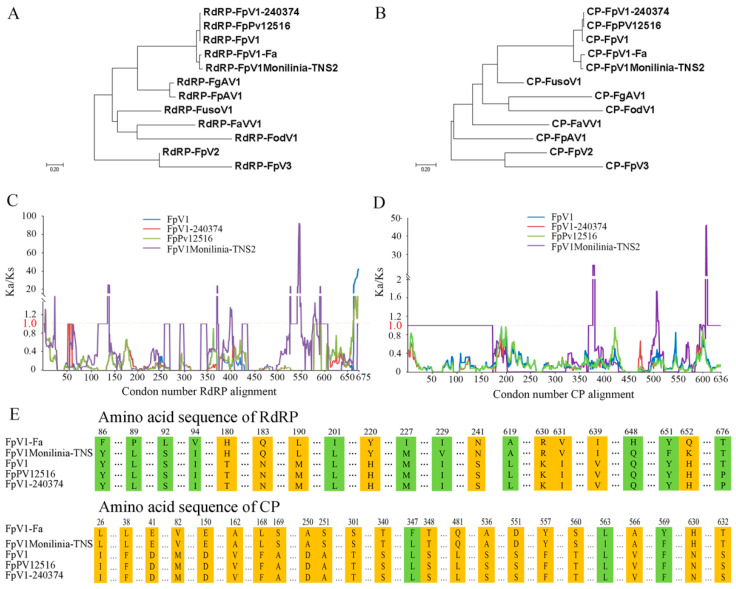
Phylogenetic analysis, sliding window analysis and amino acid sequence alignment of FpV1-Fa with other mycoviruses. Phylogenetic analysis of FpV1-Fa and related RNA viruses based on the *RdR**P* (**A**) and *CP* (**B**). The phylogenetic tree is generated by the Clustal W program. The scale bar at the lower left represents a genetic distance. The GenBank accession numbers are as follows: RdRP-FpV1Monilinia-TNS2 (MH665658.1); RdRP-FpV1 (AF047013.1); RdRP-FpPv12516 (MK279442.1); RdRP-FpV1-240374 (LC150606.1); RdRP-FusoV1 (NC_003885.1); RdRP-FaVV1 (41702310); RdRP-FgAV1 (NC_036596.1); RdRP-FpAV1 (NC_030883.1); RdRP-FpV2 (NC_030201.1); RdRP-FpV3 (27912009); RdRP-FodV1 (NC_027563.1); CP-FpV1Monilinia-TNS2 (MH665659.1); CP-FpV1-240374 (LC150607.1); CP-FpV1 (AF015924.1); CP-FpPV12516 (MK279443.1); CP-FusoV1 (NC_003886.1); CP-FaVV1 (41702309); CP-FgAV1 (NC_036601.1); CP-FpAV1 (NC_030881.1); CP-FpV2 (KU728180.1); CP-FpV3 (27912010); CP-FodV1 (NC_027565.1). Sliding window analyses selective pressures on regions surrounding evolution-guided mutagenesis. Analyses are based on pairwise alignments of RdRP (**C**) and CP (**D**) variants of FpV1-Fa and other FpV1 related mycoviruses where the ratio of non-synonymous to synonymous substitutions (K_a_/K_s_) are calculated along a 10 amino acid sliding window. The horizontal dotted line represents a K_a_/K_s_ value of 1, indicating neutrality. (**E**) Amino acid sequence alignment of FpV1-Fa with FpV1Monilinia-TNS2, FpV1, FpPv12516 and FpV1-240374. ‘…’ indicates amino acid not listed. Amino acids that changed independently in FpV1-Fa or FpV1Monilinia-TNS2 were marked in green. Amino acids that changed both in FpV1-Fa and FpV1Monilinia-TNS2 were marked in orange.

**Table 1 viruses-14-02246-t001:** Overview of *Fusarium* strains isolated during 2019 and 2021.

Sampling Sites	Fusarium Isolates Obtained	Proportion of FpV1-Fa Containing Strains (%)
Huai’an	107	13
Nanjing	60	0
Zaoyang	40	0
Linyi	80	0
Suzhou	52	0

## Data Availability

Not applicable.

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
