# Peer review of "Extending the Host Range of Fusarium Poae Virus 1 from Fusarium poae to other Fusarium Species in the Field"

_viruses, 2022, doi:10.3390/v14102246_

Round 1
Reviewer 1 Report
The manuscript entitled " Extending the host range of Fusarium poae virus 1 from Fusarium poae to other Fusarium species” reports molecular and biological characteristics of a novel strain of Fusarium poae virus 1 isolated from F. asiaticum, and named FpV1-Fa. This manuscript contains fundamental information regarding this mycovirus, including the genome sequence, phylogenic position, and biological impact on host fungus. Furthermore, the authors showed the novel FpV1-Fa can infect F. poae, F. tricinctum and F. asiaticum with high stability, and some mutations in RdRp and CP appeared to cause differences in morphology and transmission of FpV1-Fa and FpV1.
Overall, I think that this manuscript and the research presented therein is sufficient for publication in viruses with only minor modifications required.
Minor comments:
1. Result 3.1, “A phylogenetic analysis of the concatenated sequences of about 530 nt ITS was carried out.” The ITS here refers to the concatenated sequences of coupled internal transcribed spacer (ITS) and TEF1, not the internal transcribed spacer (ITS)? It's easy to misunderstand here. Please make appropriate modifications to the expression of ITS.
2. Result 3.1, “Two contigs representing the complete genomic segments of a bipartite virus were identified by RNA sequencing.” Do the authors continue with 5 'RACE and 3' RACE to verify the end sequence of the virus?If verified, you may find that the full-length sequence of the virus often has a few base differences.
3. The authors described that the previously reported FpV1 could only infect F. poae, but not other fusarium species. You haven't tested it. How do you know? Therefore, the author could change this description.
4. FpV12156 and FpPv12516 should be same.
5. For the legend of Figure 3, “-infected of”, please delete “of”.
Author Response
A list of responses to referees
We thank the two anonymous reviewers very much for helping to improve the manuscript. We have studied all the comments and have incorporated changes to all those we can in this revised manuscript. The followings are detailed point-to-point responses to the comments/suggestions made by the reviewers.
Response to Referee 1:
The manuscript entitled " Extending the host range of Fusarium poae virus 1 from Fusarium poae to other Fusarium species” reports molecular and biological characteristics of a novel strain of Fusarium poae virus 1 isolated from F. asiaticum, and named FpV1-Fa. This manuscript contains fundamental information regarding this mycovirus, including the genome sequence, phylogenic position, and biological impact on host fungus. Furthermore, the authors showed the novel FpV1-Fa can infect F. poae, F. tricinctum and F. asiaticum with high stability, and some mutations in RdRp and CP appeared to cause differences in morphology and transmission of FpV1-Fa and FpV1.
Overall, I think that this manuscript and the research presented therein is sufficient for publication in viruses with only minor modifications required.
Minor comments:
- Result 3.1, “A phylogenetic analysis of the concatenated sequences of about 530 nt ITS was carried out.” The ITS here refers to the concatenated sequences of coupled internal transcribed spacer (ITS) and TEF1, not the internal transcribed spacer (ITS)? It's easy to misunderstand here. Please make appropriate modifications to the expression of ITS.
Response: We have rewritten these sentences and modified the expression of ITS.
- Result 3.1, “Two contigs representing the complete genomic segments of a bipartite virus were identified by RNA sequencing.” Do the authors continue with 5 'RACE and 3' RACE to verify the end sequence of the virus?If verified, you may find that the full-length sequence of the virus often has a few base differences.
Response: Thanks for your kindly suggestion. We did not carry out rapid amplification of virus ends, but the RNA sequencing depth was over 2000 fold. The sequence data are close to the actual whole virus sequence including 5 'RACE and 3' RACE. We listed the sequence of 5 'RACE and 3' RACE in Figure S1.
- The authors described that the previously reported FpV1 could only infect F. poae, but not other Fusarium species. You haven't tested it. How do you know? Therefore, the author could change this description.
Response: The inappropriate descriptions have been rewritten.
- FpV12156 and FpPv12516 should be same.
Response: We correct the mistakes in the text and figures.
- For the legend of Figure 3, “-infected of”, please delete “of”.
Response: We have deleted “of” in the legend of Figure 3.

Reviewer 2 Report
Comments to the Author
The manuscript by Song et al. contains interesting findings on the host range extending of Fusarium poae virus 1 (FpV1). In this study, a novel mycovirus FpV1 (FpV1-Fa) was identified and sequenced from a new host, F. asiaticum. Unlike the FpV1 that was found previously, FpV1-Fa induced abnormal morphology and hypovirulence of F. asiaticum. FpV1-Fa can transfer from F. asiaticum to F. poae and F. tricinctum but not F. graminearum. Finally, the authors propose the possible explanation for the variation of FpV1-Fa on virulence by selection analysis. This is a very interesting study. The text, however, contains flaws that may be addressed by the authors before it can be recommended for publication.
1. Abstract
Change ‘be transformed’ to ‘transfer’.
2. Results
Please check the sentence ‘F. graminearum was the most common species of the FHB complex, comprising 65 percent of the total isolates. F. asiaticum came second, comprising 26 percent of FHB complex species.’ The phrase is inconsistent with Figure 1 which shows F. asiaticum comprises 65 percent of the total isolates.
3. Change from ‘The macroconidia was shorter in…’ to ‘The macroconidia were shorter in…’.
4. ‘…lesion length were reduced by 62%...’ ? or ‘…lesion length was reduced by 62%...’ ?
5. Figure 4. the meaning of amino acid in different color should be introduced.
6. Discussion
Change from ‘…homologous strains appeare in different regions…’ to ‘…homologous strains appeared in different regions…’.
7. Author Contributions
The phrase needs clarification. Please rewrite it.
8. References
There are some technical mistakes.
Author Response
A list of responses to referees
We thank the two anonymous reviewers very much for helping to improve the manuscript. We have studied all the comments and have incorporated changes to all those we can in this revised manuscript. The followings are detailed point-to-point responses to the comments/suggestions made by the reviewers.
Response to Referee 2:
Comments to the Author
The manuscript by Song et al. contains interesting findings on the host range extending of Fusarium poae virus 1 (FpV1). In this study, a novel mycovirus FpV1 (FpV1-Fa) was identified and sequenced from a new host, F. asiaticum. Unlike the FpV1 that was found previously, FpV1-Fa induced abnormal morphology and hypovirulence of F. asiaticum. FpV1-Fa can transfer from F. asiaticum to F. poae and F. tricinctum but not F. graminearum. Finally, the authors propose the possible explanation for the variation of FpV1-Fa on virulence by selection analysis. This is a very interesting study. The text, however, contains flaws that may be addressed by the authors before it can be recommended for publication.
- Abstract
Change ‘be transformed’ to ‘transfer’.
Response: The sentence has been revised as suggested.
- Results
Please check the sentence ‘F. graminearum was the most common species of the FHB complex, comprising 65 percent of the total isolates. F. asiaticum came second, comprising 26 percent of FHB complex species.’ The phrase is inconsistent with Figure 1 which shows F. asiaticum comprises 65 percent of the total isolates.
Response: We have corrected these mistakes in this revised manuscript.
- Change from ‘The macroconidia was shorter in…’ to ‘The macroconidia were shorter in…’.
Response: The sentence has been revised as suggested.
- ‘…lesion length were reduced by 62%...’ ? or ‘…lesion length was reduced by 62%...’ ?
Response: The sentence has been revised as ‘Seedling inoculation assays revealed that lesion length was reduced by 62% for strain Fa2021-HygB-Fpv1Fa, compared with strain Fa2021-HygB at 7 dpi’.
- Figure 4. the meaning of amino acid in different color should be introduced.
Response: We added some new sentences in Figure 3 legend to explain amino acids which marked in different color.
- Discussion
Change from ‘…homologous strains appeare in different regions…’ to ‘…homologous strains appeared in different regions…’.
Response: The sentence has been revised as suggested.
- Author Contributions
The phrase needs clarification. Please rewrite it.
Response: Author contributions have been rewritten.
- References
There are some technical mistakes.
Response: We have corrected all grammar and spell mistakes in this revised manuscript.
